# Maternal age and body mass index and risk of labor dystocia after spontaneous labor onset among nulliparous women: A clinical prediction model

Nina Olsén Nathan[1,2]*, Thomas Bergholt[3,4], Christoffer Sejling[1,5], Anne Schøjdt Ersbøll[1], Kim Ekelund[6,7], Thomas Alexander Gerds[5], Christiane Bourgin Folke Gam[8], Line Rode[9], Hanne Kristine Hegaard[1,2,4]

1 The Interdisciplinary Unit of Women's, Children's and Families' Health, the Juliane Marie Centre, Copenhagen University Hospital - Rigshospitalet, Copenhagen, Denmark, 2 Department of Obstetrics, Copenhagen University Hospital - Rigshospitalet, Copenhagen, Denmark, 3 Department of Obstetrics and Gynecology, Copenhagen University Hospital - Herlev, Herlev, Denmark, 4 Institute of Clinical Medicine, Faculty of Health Sciences and Medicine, University of Copenhagen, Copenhagen, Denmark, 5 Department of Biostatistics, University of Copenhagen, Copenhagen, Denmark, 6 Department of Anesthesia- and Operation, the Juliane Marie Centre, Copenhagen University Hospital - Rigshospitalet, Copenhagen, Denmark, 7 Copenhagen Academy of Medical Education and Simulation (CAMES), Copenhagen University Hospital–Herlev, Herlev, Denmark, 8 Department of Obstetrics and Gynecology, Nordsjællands Hospital, Hillerød, Denmark, 9 Department of Clinical Biochemistry, Copenhagen University Hospital–Rigshospitalet, Glostrup, Denmark

* nina.olsen.nathan@regionh.dk

## Abstract

### Introduction

Obstetrics research has predominantly focused on the management and identification of factors associated with labor dystocia. Despite these efforts, clinicians currently lack the necessary tools to effectively predict a woman's risk of experiencing labor dystocia. Therefore, the objective of this study was to create a predictive model for labor dystocia.

### Material and methods

The study population included nulliparous women with a single baby in the cephalic presentation in spontaneous labor at term. With a cohort-based registry design utilizing data from the Copenhagen Pregnancy Cohort and the Danish Medical Birth Registry, we included women who had given birth from 2014 to 2020 at Copenhagen University Hospital–Rigshospitalet, Denmark. Logistic regression analysis, augmented by a super learner algorithm, was employed to construct the prediction model with candidate predictors pre-selected based on clinical reasoning and existing evidence. These predictors included maternal age, pre-pregnancy body mass index, height, gestational age, physical activity, self-reported medical condition, WHO-5 score, and fertility treatment. Model performance was evaluated using the area under the receiver operating characteristics curve (AUC) for discriminative capacity and Brier score for model calibration.

**Data Availability Statement:** Our dataset originates from the Danish National Birth Registry

and is subject to legal restrictions that prohibit its sharing with other researchers. These restrictions are imposed by Danish data protection laws and regulations governing the use of sensitive health information, including data related to births and individuals' health records. As researchers, we are committed to transparency and the advancement of scientific knowledge. However, due to the legal constraints, we are unable to directly provide access to the raw data from the Danish registries. For inquiries regarding data access or further information on the legal restrictions surrounding our dataset, please contact The Research Services at The Danish Health Data Authority at kontakt@sundhedsdata.dk or +45 7221 6800. We are fully prepared to facilitate the sharing of aggregated results, summary statistics, and any analysis code used in our study. Additionally, we are more than willing to collaborate with interested parties to explore avenues for replication or further investigation within the boundaries of Danish data protection laws.

**Funding:** This study (author NON) was supported by Copenhagen University Hospital – Rigshospitalet's Research grant. Hans og Nora Buchards Fond (grant no Journal nr. 7334, ID 1894) provided funding for publication. The funders had no role in study design, data collection and analysis, decision to publish, or preparation of the manuscript.

**Competing interests:** The authors have declared that no competing interests exist.

## Results

A total of 12,445 women involving 5,525 events of labor dystocia (44%) were included. All candidate predictors were retained in the final model, which demonstrated discriminative ability with an AUC of 62.3% (95% CI:60.7–64.0) and Brier score of 0.24.

## Conclusions

Our model represents an initial advancement in the prediction of labor dystocia utilizing readily available information obtainable upon admission in active labor. As a next step further model development and external testing across other populations is warranted. With time a well-performing model may be a step towards facilitating risk stratification and the development of a user-friendly online tool for clinicians.

## Introduction

Labor dystocia, also referred to as prolonged labor, is defined by slow progression of cervical dilation in the first stage of labor and/or fetal descent in the second stage [1, 2]. Being common among nulliparous women, affecting as many as 20–47% in the Scandinavian countries [3–6], labor dystocia remains a challenge for women in labor, midwives, and obstetricians.

Labor dystocia is a major indication for instrumental vaginal delivery and cesarean section [3, 7, 8]. It is also a key contributor to women's negative birth experiences [9–11]. Additionally, although synthetic oxytocin is extensively used for labor augmentation [12, 13] it is important to acknowledge that the Institute For Safe Medication Practices has classified the drug as one of twelve high-alert medications signifying a heightened risk of imposing serious acidotic harm to the fetus due to risk of hyperstimulation of the feto-placenta entity [14].

The current emphasis in obstetrics is to find better ways to diagnose and treat labor dystocia, and until now research has primarily focused on either management [12, 15–17] or identification of factors associated with labor dystocia [18–22]. Still, clinicians have no tools to appropriately risk-stratify women at the onset of labor. Generally, in medicine, prediction models provide individualized risk estimates for clinically important outcomes in patients with a particular characteristic, disease, or condition. Prediction models can provide more accurate prognoses than clinicians working on their own [23]. In recent years, risk prediction models have been developed in feto-maternal medicine [24–27] and implemented in daily clinical practice [28]. In comparison, prediction models in the field of obstetrics are less prevalent and are often related to predicting mode of birth [29–32]. Notably, two recently published studies from Japan and China have sought to predict the risk of delivery by emergency caesarean section due to labor dystocia [33, 34]. To our knowledge, no prediction model estimating a woman's risk of labor dystocia at time of labor onset exists.

Identification of women with increased risk of labor dystocia could have several benefits. First, a prediction model would provide an individualized risk assessment incorporating multiple predictor variables while considering their combined effect. Second, a well-performing model could be further developed into an online calculator and serve as a supportive decision tool for pregnant women and clinicians, enabling them to implement preventive measures to potentially mitigate the risk.

The aim of the study is to develop a prediction model for labor dystocia among nulliparous women in spontaneous labor at term with a single baby in cephalic presentation to be used at onset of active labor.

## Material and methods

### Population

The source population for this study was derived from Copenhagen University Hospital–Rigshospitalet. Since 2012 pregnant women have been sent an electronic clinical questionnaire when booking their combined first-trimester screening for chromosomal anomalies. More than 90% of all pregnant women in Denmark participate in this screening [35]. The questionnaire collects self-reported data related to socio-demographic characteristics, medical and obstetric history, mental health and lifestyle factors [36, 37] and this is transferred to the electronic medical journal with the aim of qualifying and optimizing antenatal care. When researchers, with approvals from relevant authorities, utilize anonymized data from this questionnaire it is cited as Copenhagen Pregnancy Cohort.

Because all persons in Denmark have a unique identity number unambiguous linkage of data from Copenhagen Pregnancy Cohort with corresponding birth data in the Danish Medical Birth Registry (MBR) was possible. From the period October 2012 to December 2020 39,162 questionnaires were sent and the response rate was 89% (n = 34,883). After removing pregnancies not eligible for the study (i.e., miscarried before questionnaire completion or self-reported multiparous pregnancies) 20,271 pregnancies remained for linkage.

For obtaining the target population the following stepwise exclusion criteria were applied to births identified in MBR: stillbirth, multipara, preterm birth (gestational age $\leq 36^{+6}$ weeks), multiple gestations, induction of labor or elective caesarean section, non-vertex cephalic or breech presentations, and year of birth before 2014, as this was the year the Danish National Clinical Guideline concerning labor dystocia was published [1]. Additionally, if discrepancies were identified between due date from Copenhagen Pregnancy Cohort and in MBR, the birth was not included. Subsequently, 12,717 nulliparous women with spontaneous onset of labor at term with a cephalic presentation constituted the target population. Due to incomplete data an additional 272 births (2.1%) were excluded resulting in a study population of 12,445 (Fig 1).

### Selection of predictors and classification of outcome

Eight candidate predictors were pre-selected based on clinical reasoning and established associations in the literature [8, 18, 20, 22, 38–40]. Inferential associations or causality reflections were not a requirement as the goal was to optimize predictive accuracy and not explain causality between variables. There was no targeted maximum number of variables but a prespecified aim to achieve a model applicable to clinical practice guided the process. The predictors included were maternal age at time of birth in (years, continuous), pre-pregnancy body mass index (kg/m$^2$, continuous), height ($\leq 160$ cm and >160 cm), gestational age (weeks + days, $37^{+0}$–$37^{+6}$, $38^{+0}$–$38^{+6}$, $39^{+0}$–$39^{+6}$, $40^{+0}$–$40^{+6}$, $\geq 41^{+0}$), weekly physical activity in 1st trimester (none, 1–3.5 hours, $\geq 3.5$ hours) [41]. In relation to general and reproductive health, we included self-reported medical conditions (none, somatic, psychiatric, both somatic and psychiatric), WHO-5 Well-being Index (score of $\leq 50$, >50) [42], and fertility treatment (yes/no). We omitted smoking and alcohol consumption during pregnancy as the prevalence was very low (2%).

The outcome we sought to predict was labor dystocia derived from MBR–in which coding is based on the Danish Healthcare Classification System including the International Classification of Disease version 10 [43]. In the period up to 2019 treatment with oxytocin augmentation

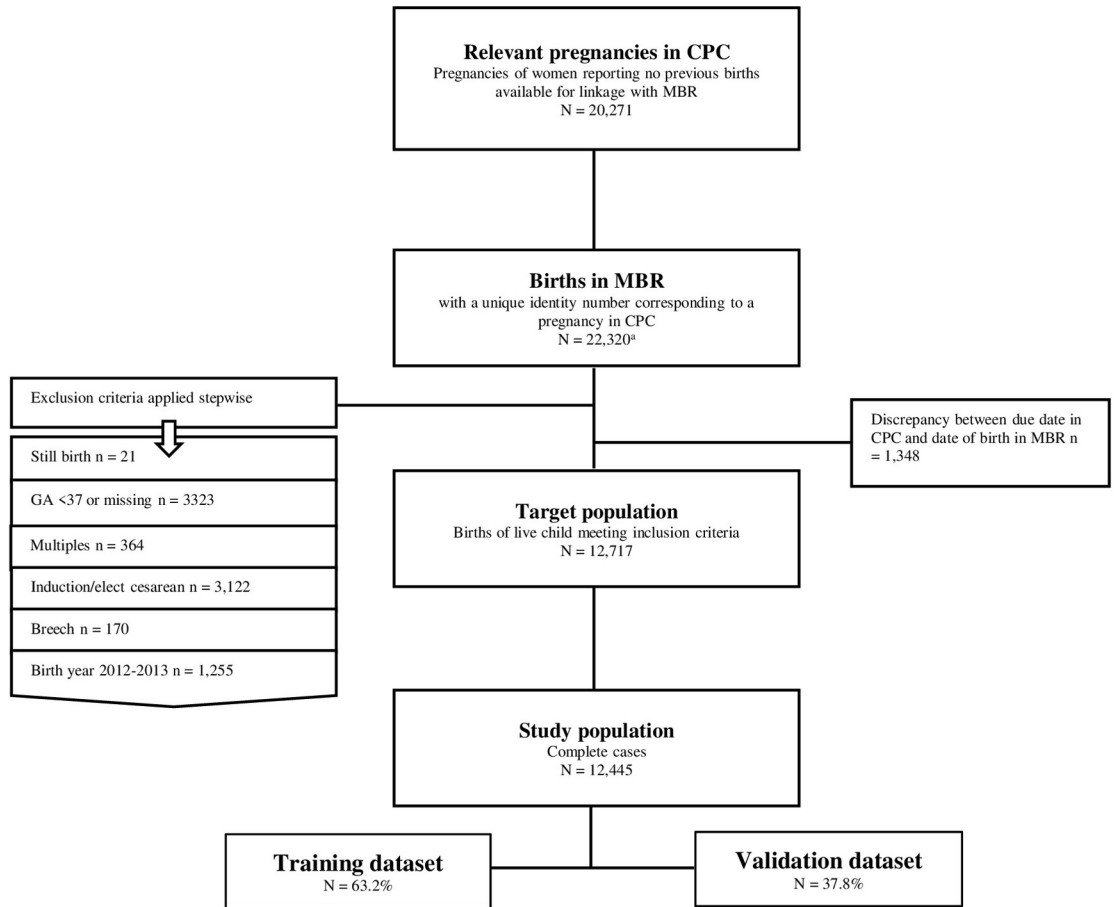

**Fig 1. Flowchart of study population derived from Copenhagen Pregnancy Cohort and linked with corresponding birth data from the Danish Medical Birth Registry.** Abbreviations: CPC, Copenhagen Pregnancy Cohort; MBR, the Danish Medical Birth Registry; GA, gestational age. [a]A woman from CPC could have more than one pregnancy in MBR.

was available, while the diagnosis code 'O62*: obstructed labor' was available from 2019 and onward. Supporting information presents all predictor candidates and outcome with defini-tion/diagnostic codes, measurement scale, and source of data (S1 Table).

## Statistical analysis

The study population was split randomly [44], resulting in a training dataset of 63.2% of the complete cases (no missing values) and a validation dataset of the remaining 36.8% of the complete cases. Population characteristics were presented as medians with interquartile ranges and frequencies with percentages as appropriate. We pre-specified a list of prediction models based on logistic regression (Table 1). Utilizing the training dataset, a discrete super learner—an algorithm designed to ease the challenge of choosing the most suitable predictive model by considering various strategies—chose the logistic regression model from the pre-specified list with the lowest 10-fold cross-validated prediction performance (Brier score) [44]. The Brier score is the squared difference between the predicted probability of labor dystocia and the observed status, the lower the Brier score the higher the predictive performance. The validation dataset was used to calculate the receiver operating characteristic (ROC) curve, the calibration plot, the area under the ROC curve (AUC), and the Brier score of the selected super learner

**Table 1. Candidate prediction models.**

Candidate prediction models for outcome labor dystocia built by logistic regression using the following candidate predictor variables:
Category I: maternal age, body mass index (BMI), height, gestational age (GA), physical activity
Category II: medical condition disease, WHO-5 score, fertility treatment

| | |
|---|---|
| **Model 1** | Additive effects of category I variables, assumed linear relationship between linear predictor (logit odds) and maternal age, BMI |
| **Model 2** | Additive effects of category I, restricted cubic splines model the relationship for maternal age, BMI |
| **Model 3** | Alike Model 1, but with statistical interactions, allowing the effects of the variables maternal age, BMI, GA to depend on the three categories defined physical activity |
| **Model 4** | Alike Model 2, but with statistical interactions, allowing the effects of the variables maternal age, BMI, GA to depend on the three categories defined physical activity |
| **Model 5** | Alike Model 1 now adding additive effects of all category II variables |
| **Model 6**[a] | Alike Model 2 now adding additive effects of all category II variables |
| **Model 7** | Alike Model 3 now adding additive effects of all category II variables. |
| **Model 8** | Alike Model 4 now adding additive effects of all category II variables |

[a]The final model selected, based on best performance (Brier score S1 Fig), tested in validation dataset

model. Presented are personalized risk predictions from the super learner model. The statistical analysis was carried out with R [45].

### Ethics statement

The study was approved by the National Data Protection Agency (journal no.: RH-2016-202, I-Suite no.: 04778). Patient consent was waived due to the Danish Patient Safety Authorities having granted permission to disclose patient information from medical registries for research use (journal no.: H-21032059).

## Results

The final cohort included 12,445 births involving 5,525 women with labor dystocia (44%). The mean age of women was 30.1 years and 87% were of Danish nationality. Median gestational age at birth was 283 days (40+3). Mode of delivery was distributed with 77% vaginal birth, 14% instrumental birth and 9% caesarean birth. Maternal characteristics and predictors are presented in Table 2.

### Model development

The discriminative ability for all candidate models was better than of a benchmark null model that ignores predictor candidates and predicts an average risk. Brier score was not substantially different across models (S1 Fig). In the best performing selected model, model six, all candidate predictors were retained, and restricted cubic splines were used for the continuous candidate predictors age and BMI (Table 1). Regression coefficients for the predictors in the selected model are available in supplementary material (S2 Table).

### Internal validation and model performance

We inspected the performance of the selected model in the validation dataset on a calibration plot (Fig 2). When the predicted risk of labor dystocia was compared with the observed rate, we saw a well calibrated model for all risk strata. The calculated AUC for the ability of the

**Table 2. Characteristics for study population.**

| | Training dataset | Validation dataset | Total |
|---|---|---|---|
| | **N = 7,866** | **N = 4,579** | **N = 12,445** |
| **General characteristics** | | | |
| **Maternal age at birth** years, median (IQR) | 30.2 (28–33) | 30 (28–33) | 30.1 (28–33) |
| **Nationality** n (%) | | | |
| Danish | 6,743 (87.2) | 3,901 (86.5) | 10,644 (86.9) |
| other | 987 (12.8) | 611 (13.5) | 1,598 (13.1) |
| missing | 136 | 67 | 203 |
| **Cohabitating** n (%) | | | |
| Yes | 7,232 (92.2) | 4,149 (90.9) | 11,381 (91.8) |
| No | 609 (7.8) | 413 (9.1) | 1,022 (8.2) |
| missing | 25 | 17 | 42 |
| **Level of education** n (%) | | | |
| Compulsory | 511 (6.6) | 311 (7.0) | 822 (6.8) |
| Skilled | 228 (3.0) | 126 (2.8) | 354 (2.9) |
| Tertiary (1–2 years) | 382 (5.0) | 241 (5.4) | 623 (5.1) |
| Bachelor or equivalent (3–4 years) | 2,270 (29.5) | 1,286 (28.8) | 3,556 (29.2) |
| Master or equivalent ($\geq$5 years) | 4,298 (55.9) | 2,507 (56.1) | 6,805 (56.0) |
| missing | 177 | 108 | 285 |
| **Height** cm, median (IQR) | 169 (165–173) | 168 (164–173) | 169 (165–173) |
| >160 | 825 (10.5) | 498 (10.9) | 1,323 (10.6) |
| $\leq$160 | 7,041 (89.5) | 4,081 (89.1) | 11,122 (89.4) |
| **Pre-pregnancy BMI** kg/m2, median (IQR) | 21.7 [20.2, 23.8] | 21.8 [20.2, 23.8] | 21.8 [20.2, 23.8] |
| **Health** | | | |
| **WHO-5 Well-being Index score** median (IQR) | 64 (52–72) | 64 (52–72) | 64 (52–72) |
| >50 | 6,335 (80.5) | 3,733 (81.5) | 10,068 (80.9) |
| $\leq$50[a] | 1,531 (19.5) | 846 (18.5) | 2,377 (19.1) |
| **Medical condition** n (%) | | | |
| None | 7,073 (89.9) | 4,101 (89.6) | 11,174 (89.8) |
| Somatic | 572 (7.3) | 326 (7.1) | 898 (7.2) |
| Psychiatric | 192 (2.4) | 136 (3.0) | 328 (2.6) |
| Somatic and psychiatric | 29 (0.4) | 16 (0.3) | 45 (0.4) |
| **Lifestyle** | | | |
| **Physical activity** hours weekly, n (%) | | | |
| None | 2,941 (37.4) | 1,770 (38.7) | 4,711 (37.9) |
| <3.5 | 1,064 (13.5) | 580 (12.7) | 1,644 (13.2) |
| $\geq$3.5[b] | 3,861 (49.1) | 2,229 (48.7) | 6,090 (48.9) |
| **Obstetric characteristics** | | | |
| **Fertility treatment** n (%) | | | |
| Yes | 1,014 (12.9) | 579 (12.6) | 1,593 (12.8) |
| No | 6,852 (87.1) | 4,000 (87.4) | 10,852 (87.2) |
| **Gestational age at birth** days, median (IQR) | 283 (277–288) | 283 (277–288) | 283 (277–288) |
| weeks + days, n (%) | | | |
| 37+0–37+6 | 361 (4.6) | 183 (4.0) | 544 (4.4) |
| 38+0–38+6 | 750 (9.5) | 443 (9.7) | 1,193 (9.6) |
| 39+0–39+6 | 1,642 (20.9) | 893 (19.5) | 2,535 (20.4) |
| 40+0–40+6 | 2,686 (34.1) | 1,532 (33.5) | 4,218 (33.9) |
| $\geq$41+0 | 2,427 (30.9) | 1,528 (33.4) | 3,955 (31.8) |

*(Continued)*

**Table 2.** (Continued)

| | Training dataset | Validation dataset | Total |
|---|---|---|---|
| | **N = 7,866** | **N = 4,579** | **N = 12,445** |
| **Labor dystocia** n (%) | | | |
| Yes | 3470 (44.1) | 2055 (44.9) | 5525 (44.4) |
| No | 4396 (55.9) | 2524 (55.1) | 6920 (55.6) |
| **Mode of birth** n (%) | | | |
| Vaginal | 6,033 (77.0) | 3,479 (76.3) | 9,512 (76.7) |
| Instrumental | 1,094 (14.0) | 643 (14.1) | 1,737 (14.0) |
| Cesarean section | 713 (9.0) | 438 (9.6) | 1,151 (9.3) |
| missing | 26 | 19 | 45 |

Abbreviation: BMI, body mass index

[a]Indicates reduced psychological well-being

[b]Indicates adherence with national recommendations

model to predict labor dystocia was 62.3% (95% CI:60.7–64.0) indicating poor discrimination (Fig 3). Brier score was 0.24 and corresponding Index of Predictive Accuracy 0.046.

## Risk scenarios

To illustrate the impact of each predictor we modelled three scenarios–a low-, reference, and high-risk (Table 3). In the low-risk scenario, a 20-year-old woman giving birth in gestational week 40, with BMI of 20 kg/m$^2$, height >160 cm, no medical conditions, physically active ≥3.5 hours weekly, reporting a WHO-5 score of >50 and having received fertility treatment had a predicted risk of labor dystocia of 24%. Conversely in the high-risk scenario a 40-year-old woman in gestational week 41 or later, with a BMI of 35 kg/m$^2$, height ≤160 cm, somatic medical condition, not physically active, reporting low well-being defined as a WHO-5 score of ≤50 and having conceived spontaneously had a prediction risk of 88%. For a selected overview

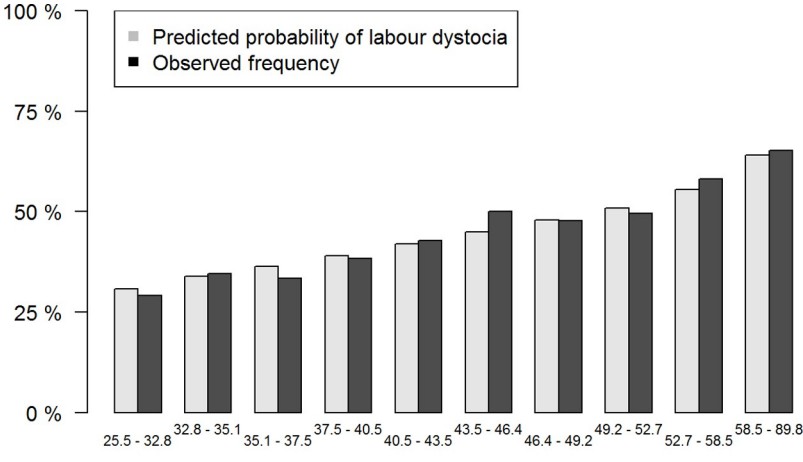

**Fig 2. Calibration plot showing the agreement between predicted probabilities and observed frequency of labor dystocia in the validation data grouped according to deciles of predicted risk of the selected model.**

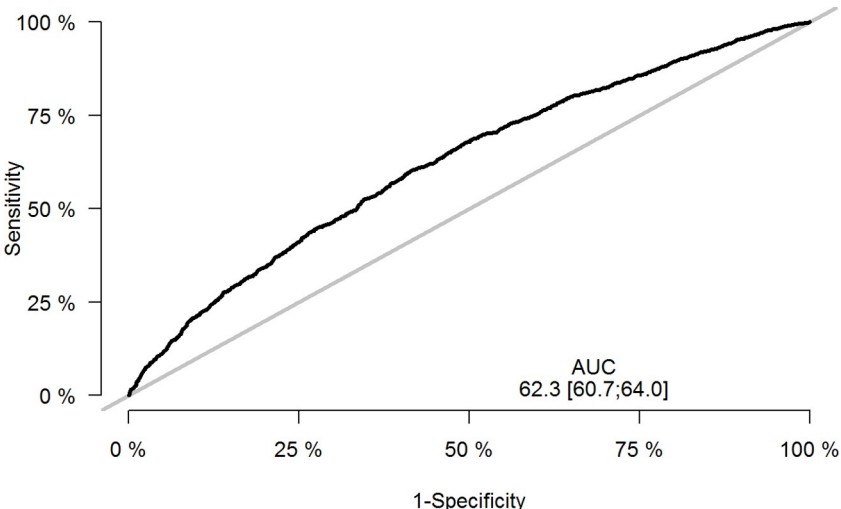

**Fig 3. Receiver operating characteristic (ROC) curve with calculated area under the curve (AUC) in the validation dataset.**

of personalized predicted risk for labor dystocia with unique combinations of maternal age and BMI, see supporting information (S3 Table) [46].

Also, in the supporting information a figure of the running risk as a function for sequentially 'switching-on' categorical predictors is available (S2 Fig).

### Maternal age and BMI

Evidently increasing maternal age and BMI were strong predictors. Specifically, for these we show the span of predicted risks when restraining all other predictors as described in the previous scenarios (Fig 4). We noted an increasing predicted risk by increasing age and BMI for all three scenarios. Exemplified a 40-year-old woman demonstrated a two-fold increase in her projected risk compared to a 20-year-old also in the low-risk scenario.

### Discussion

In this cohort-based registry study, we developed a prediction model for labor dystocia among nulliparous women in spontaneous term labor with a singleton baby in cephalic presentation. The selected model included all eight candidate predictors, and the predicted risk of labor dystocia was higher with increasing age and BMI and substantially higher among women with all predictors. The predictive performance was poor with Brier score of 24% and AUC of 63%.

The finding that pre-pregnancy BMI has predictive value is supported by previous epidemiological studies showing a relationship between obesity and length of labor [47–49]. There are

**Table 3. Predicted risk for three scenarios.**

| Scenario | Age years | BMI kg/m² | GA weeks +days | Physical activity hours weekly | Height cm | Medical condition | Fertility treatment | WHO-5 score | Predicted risk % |
|---|---|---|---|---|---|---|---|---|---|
| **Low-risk** | 20 | 20 | 40+0–40+6 | ≥3.5 | >160 | none | yes | >50 | 24.0 |
| **Reference** | 30 | 27 | 40+0–40+6 | ≥3.5 | >160 | none | no | >50 | 41.7 |
| **High-risk** | 40 | 35 | ≥41+0 | none | ≤160 | somatic | no | ≤50 | 88.1 |

Abbreviations: BMI, body mass index; GA, gestational age

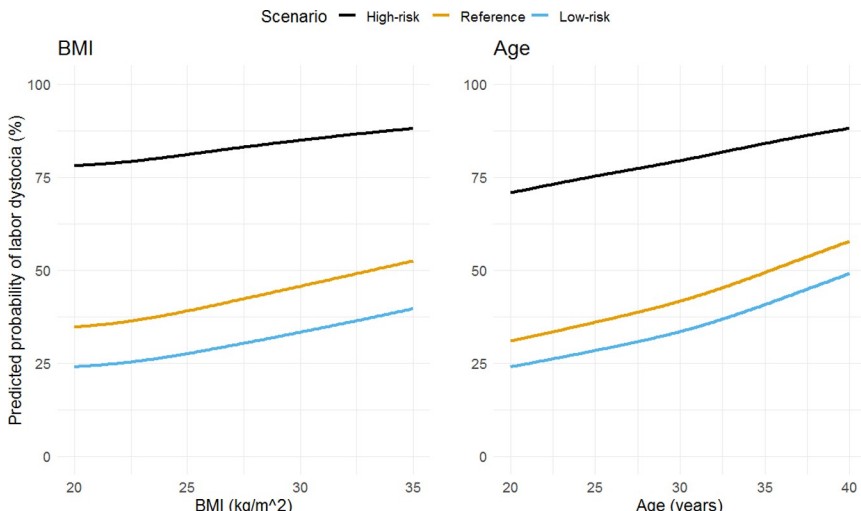

**Fig 4. Predicted risk for three scenarios allowing change in maternal BMI and age.** Abbreviations: BMI, body mass index. [a]All predictors, other than BMI and age, are retained as described in Table 3.

indications that obese pregnant women at term have cellular alterations in the uterine myocytes which could decrease contractility [50–52]. Also the association between a women's age and labor dystocia is well-established, with evidence suggesting a negative effect of ageing [20, 22, 53]. In concordance with our model a recent study, aimed at developing a prediction model for caesarean section due labor dystocia after induction of labor, found age to be an important predictor [32]. Still, the explanatory mechanism is not known and in vitro studies of human uterine myocyte contractility have failed to demonstrate reduced contractility associated with age [54], as such validated physiological explanations have yet to be found.

An unexpected finding was that *not* receiving fertility treatment was a predictor for labor dystocia. This was contrary to the expected direction of the effect as existing evidence has found that pregnancies conceived after fertility treatment are at increased risk for obstetric complications [55, 56]. The finding could potentially be attributed to the 'healthy-user effect', wherein women who receive fertility treatment engage in a healthier diet and lifestyle compared to those who conceive spontaneously. The available evidence supporting this explanation is however lacking and a previous study conducted in the Copenhagen Pregnancy Cohort population found that fertility treated women were less likely to engage in physical activity [57]. Another possible explanation might be that the least healthy women receiving fertility treatment were excluded due to preterm birth and induction of labor in relation to the preliminary steps of identifying the target population in the present study.

To develop a more robust and accurate prediction model for labor complications, future models could consider a broader range of relevant predictors to capture the complexity of the underlying mechanisms and enhance model performance. The occurrence of labor dystocia is likely attributed to a complex interplay of numerous contributing factors and intrapartum events undoubtedly play a role. Furthermore, it is probable that there is a genetic component to labor dystocia, and studies have found an increased risk in women whose mothers or sisters also experienced the complication [58, 59]. Still, calculations of heritability leave the majority of labor dystocia with no genetic cause [59].

In summary, several complex factors will affect labor dystocia, and the complication requires a comprehensive approach. Nonetheless, the aim of this model was not to inform decisions regarding mode of birth or influence the clinical treatment of labor dystocia, but to

serve as a first step in developing a model that can support risk stratification of women at the onset of labor to target resources and facilitate discussion of prevention. Though evidence on optimal prevention is not straightforward there are studies that support certain practices and interventions. Avoidance of hospital admission in the latent stage of labor may reduce the risk of receiving augmentation [60–62]. Also the utilization of continuous labor support has been associated with a reduction in the incidence [63]. While adequate hydration, specifically intravenously administrated, has provided conflicting evidence [64]. It is the position of the authors that prevention and stepwise progression of interventions aimed at women at high risk is the preferred mode of action.

A central strength of this study is the large sample size and the number of events observed. Another strength is that the model's source population comes from at large cohort with a high coverage combined with national registry-based data. The latter ensuring a representative sample of the target population and limiting non-response and loss to follow-up. We pre-selected candidate predictors based on available evidence and clinical reasoning instead of observed significant relationships with outcome in the dataset, thereby ensuring higher external validity and less overfitting of the model [44]. Furthermore, we chose only to include predictor candidates present at the onset of labor, enabling the model's applicability in current labor care.

There are also limitations. Firstly, and importantly, is the lack of external validation for the prediction model, as it was solely internally validated. Given the unique demographic characteristics of our population, the low levels of smoking and alcohol consumption and high level of education, it is crucial to recognize that the performance of the model may vary when applied to other populations. This is also true for the candidate predictors the model has been explicitly trained to consider and model performance is heavily influenced by for example the BMI distribution in the Copenhagen Pregnancy Cohort study. Another key limitation of MBR in the period 2014–2019 is the lack of diagnostic codes related to labor complications. Using the proxy of treatment by oxytocin augmentation entails the risk of introducing misclassification bias, occurring if oxytocin augmentation does not consistently align with the presence of labor dystocia. Though diagnosis and treatment are used interchangeably in many publications [16, 65, 66], no studies have validated the accuracy of oxytocin augmentation as a measure of labor dystocia in the Danish registries [67]. It is, however, reassuring that publicly accessible aggregated data shows the prevalence of the labor dystocia diagnosis is consistent with the prevalence in our data and has remained relatively stable since 2014 [68]. Furthermore, the distribution of the outcome is roughly the same across the study period (S3 Fig).

Most candidate predictors were patient-reported outcome data (PRO-data), and this has enabled the model to cover a broad range of domains on which detailed information is not available in the standard registries, but would be obtainable by clinicians, like hours of physical activity per week. On the other hand, the data was collected in the first trimester, and changes across pregnancy may be significant. One study found that while a pre-pregnancy BMI was an independent risk factor, the risk of dystocia increased as a function of gestational weight gain [69]. Additionally, intrapartum actions and interventions may also predict the risk of labor dystocia but were unavailable for incorporation in the model. Neither were we able to subclassify labor dystocia according to the phase of labor This distinction could have a clinically relevant implication, as research suggests that the mode of birth is influenced by the timing of labor dystocia diagnosis. Specifically, women diagnosed at ≤5 cm of cervical dilation exhibit higher cesarean section rates [70]. However, obtaining information on the phase of labor from MBR is not possible. While a robust prediction model requires a large population; making the optimal data source a national registry, this approach also poses a limitation due to national registries lacking the additional clinical details sometimes desired.

We applied a single-random split to our dataset to account for potential temporal changes due to revisions of the MBR or clinical guideline updates, but introducing randomness can inadvertently influence the performance of the prediction model [44]. Alternatively, with a calendar year split we would have avoided this while mimicking real life where data from women in the past is used to build a model applicable for future use.

Another methodological choice was to perform a complete case analysis, excluding observations of with missing values. Hereby we could maintain the original relationship and patterns in the complete dataset and avoid introducing assumptions or imputed values that potentially may affect the relationship. In contrast, we acknowledge the loss of information when discarding observations in the model.

## Conclusion

The present model serves as a first step in predicting labor dystocia using information obtainable upon admission. Maternal age and BMI were found to be important predictors influencing nulliparous women's risk of labor dystocia. The model's prediction ability was reasonable, but not satisfactory for clinical application. As with any new model, there is a need for further development, extensive testing, and external validation in other settings to ensure that the model performs well in different populations or clinical settings. We hope other researchers will build upon our work - ultimately advancing the field towards more accurate predictive models. Over time, as the development of robust models for predicting labor dystocia progresses, creating a user-friendly online calculator tool, primarily aimed at clinicians, emerges as a logical next step.

## Supporting information

**S1 Table. Definition and description of predictors and outcome.**
(PDF)

**S2 Table. Selected model input with associated coefficients and standard error.**
(PDF)

**S3 Table. Predicted risk for labor dystocia for selected combinations of age and BMI.**
(PDF)

**S1 Fig. Brier scores for each candidate model at different ridge penalty values–model selection.** Abbreviation: m, model. Model six was the final selected model for which model performance was tested in validation dataset. See Table 1 for specifications for all eight candidate models.
(TIF)

**S2 Fig. Switch-on risk development.** The figure shows the effect of 'switching on' exposures starting from 'None' which is the low-risk scenario defined in Table 3. Only categorical candidate predictors are incorporated in the figure. The order is as follows: Gestational age from 40 +0–40+6 to ≥41+0; Height from >160cm to ≤160cm; Physical activity from ≥3.5 hours weekly to no physical activity; Fertility treatment from yes to no; Medical condition from none to somatic; WHO-5 score from >50 to ≤50.
(TIF)

**S3 Fig. Distribution of outcome across period of study (2014–2020).** In the years 2014–2018, the outcome is defined by oxytocin augmentation, while from year 2019 the outcome is defined by ICD-10 diagnosis codes.
(TIF)

## Acknowledgments

Camilla Byskou Eriksen substantially contributed to data preparation of the Copenhagen Pregnancy Cohort in the early stages of project.

## Author Contributions

**Conceptualization:** Nina Olsén Nathan, Thomas Bergholt, Anne Schøjdt Ersbøll, Kim Ekelund, Christiane Bourgin Folke Gam, Line Rode, Hanne Kristine Hegaard.

**Data curation:** Nina Olsén Nathan, Line Rode, Hanne Kristine Hegaard.

**Formal analysis:** Nina Olsén Nathan, Christoffer Sejling, Thomas Alexander Gerds.

**Funding acquisition:** Nina Olsén Nathan, Hanne Kristine Hegaard.

**Investigation:** Nina Olsén Nathan, Thomas Alexander Gerds.

**Methodology:** Nina Olsén Nathan, Christoffer Sejling, Thomas Alexander Gerds.

**Project administration:** Nina Olsén Nathan.

**Supervision:** Thomas Bergholt, Kim Ekelund, Thomas Alexander Gerds, Line Rode, Hanne Kristine Hegaard.

**Visualization:** Nina Olsén Nathan, Thomas Bergholt, Christoffer Sejling.

**Writing – original draft:** Nina Olsén Nathan.

**Writing – review & editing:** Nina Olsén Nathan, Thomas Bergholt, Christoffer Sejling, Anne Schøjdt Ersbøll, Kim Ekelund, Thomas Alexander Gerds, Christiane Bourgin Folke Gam, Line Rode, Hanne Kristine Hegaard.

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
