## [Decision Letter · Decision Letter 0]

26 Apr 2024

PONE-D-24-07911Maternal age and body mass index and risk of labor dystocia after spontaneous labor onset among nulliparous women: A clinical prediction modelPLOS ONE

Dear Dr. Nathan,

Thank you for submitting your manuscript to PLOS ONE. After careful consideration, we feel that it has merit but does not fully meet PLOS ONE’s publication criteria as it currently stands. Therefore, we invite you to submit a revised version of the manuscript that addresses the points raised during the review process.

We look forward to receiving your revised manuscript.

Kind regards,

David Desseauve, MD, MPH, PhD

Academic Editor

PLOS ONE

“This study (author NON) was supported by Copenhagen University Hospital – Rigshospitalet’s Research grant. Hans og Nora Buchards Fond provided funding for publication.”

Reviewers' comments:

Reviewer's Responses to Questions

**Comments to the Author**

1. Is the manuscript technically sound, and do the data support the conclusions?

Reviewer #1: Yes

Reviewer #2: No

Reviewer #3: Yes

2. Has the statistical analysis been performed appropriately and rigorously? 

Reviewer #1: Yes

Reviewer #2: I Don't Know

Reviewer #3: Yes

3. Have the authors made all data underlying the findings in their manuscript fully available?

Reviewer #1: Yes

Reviewer #2: No

Reviewer #3: Yes

4. Is the manuscript presented in an intelligible fashion and written in standard English?

Reviewer #1: Yes

Reviewer #2: Yes

Reviewer #3: Yes

5. Review Comments to the Author

Reviewer #1: Thank you for giving me the opportunity to read this paper. The issue of labor dystocia is of real interest in obstetrics. Any serious work that can help the obstetrician in this field is important.

The authors describe their objective and how they achieved it. They also highlight the limitations of the study, which are mostly the lack of external validation limiting the application of these results to other populations.

One of its strengths is that it uses patient centered indicators such as the WHO well-being score and self reported outcomes.

I found this work to be well written and well explained, with a great deal of thought given to both the value of this work and its current and future limitations.

I would like to thank the authors for their work.

NB : Flow chart is not readable.

Reviewer #2: I thank the authors for their work, which I found interesting. Indeed, it would be interesting to have a model that predicts the risk of dystocia and more importantly to predict whether therapeutic measures are efficient (ie. worth the wait and the risk of oxytocin augmentation)

Nevertheless, I feel that several modifications need to be made in the design of the study to assess this important problematic.

First and mostly: the outcome to be predicted.

1. The authors took labour dystocia, without differentiating dystocia during the active phase / descent phase or expulsion phase, which would potentially have different impact, namely a caesarean section is more likely in case of dystocia during the active phase than during the expulsive phase, where an instrumental delivery would be possible. The consequences for maternal morbidity are different.

2. they choose to predict all dystocia, without taking into account the measures taken ie. amniotomy or oxytocin augmentation, at least after 2019 there is no information, and therefore I find it difficult to draw conclusions on whether the dystocia was resolved or not, and more importantly whether it was associated with an “adverse” outcome such as caesarean section, instrumental delivery.

Second :I find it difficult to understand who was the training dataset and the validating dataset chosen. How many cases of dystocia were in each data set. Figure 1 is unreadable

Third : I feel that the variables testing should be presented as main results. It is indeed the whole point of the study and should not be presented as supplementary data.

Finally In the end their model seems to have a poor discrimination value…

Reviewer #3: To sum up, the study is interesting. A bibliography review shows a few articles on the subject, confirming the review you have already carried out on the subject. The idea of including only predictors present at the onset of labor is very interesting, so we can also screen for them upstream and use them clinically.

The statistical method used in the study seems rigorous and appropriate for developing a predictive model.

It's worth noting that the model's predictive performance is currently modest, with a Brier score of 24% and an AUC of 63%. However, this also presents an opportunity for future research to strengthen its predictive capabilities.

This article is only a first step, but it makes us think about the risk factors for labor dystocia.

The summary is well written.

It respects the form and content of the article.

Regarding the introduction, it leads to an apparent problem by showing the benefits of prediction scores.

The results of fertility treatments are interesting. Whether they are confirmed or just the result of chance (or a confounding factor) remains to be seen.

6. PLOS authors have the option to publish the peer review history of their article (what does this mean?). If published, this will include your full peer review and any attached files.

Reviewer #1: No

Reviewer #2: No

Reviewer #3: **Yes: **Buisson Alexandre

---

## [Author Response · Author response to Decision Letter 0]

10 May 2024

Manuscript PONE-D-24-07911

‘Maternal age and body mass index and risk of labor dystocia after spontaneous labor onset among nulliparous women: A clinical prediction model’

Dear Editor David Desseauve and peer-reviewers,

We want to thank you very much for your consideration of our manuscript and suggested revisions. We appreciate the points raised, which have helped improve the manuscript considerably. Please see our point-by-point response to the comments below in red font. The yes/no answers pertaining to questions (Q) 1-4 and 6 are discussed when relevant together with the reviewer’s detailed comments under question 5. 

Comments to the Authors

Q1. Is the manuscript technically sound, and do the data support the conclusions?

Reviewer #1 Yes

Reviewer #2 No

Reviewer #3 Yes

Q2. Has the statistical analysis been performed appropriately and rigorously? 

Reviewer #1 Yes

Reviewer #2 I Don’t Know

Reviewer #3 Yes

Q3. Have the authors made all data underlying the findings in their manuscript fully available?

Reviewer #1 Yes

Reviewer #2 No

Reviewer #3 Yes

Q4. Is the manuscript presented in an intelligible fashion and written in standard English?

Reviewer #1 Yes

Reviewer #2 Yes

Reviewer #3 Yes

Q5. Review Comments to the Author

Reviewer #1: Thank you for giving me the opportunity to read this paper. The issue of labor dystocia is of real interest in obstetrics. Any serious work that can help the obstetrician in this field is important.

Thank you for acknowledging the importance of labor dystocia, the focus of our prediction model. Indeed, we authors undertook this work to support obstetricians through research aimed at qualifying the management of this prevalent labor complication. Your assessment of the manuscript as relevant and technically sound with supported conclusions (Q1), statistically rigorous (Q2), with data underlying the findings (Q3), and effectively disseminated (Q4) is greatly appreciated. 

The authors describe their objective and how they achieved it. They also highlight the limitations of the study, which are mostly the lack of external validation limiting the application of these results to other populations. 

One of its strengths is that it uses patient centered indicators such as the WHO well-being score and self reported outcomes.

As noted by Reviewer 1 and addressed in the manuscript's discussion section, a limitation of this study is the lack of external validation (L287). This was anticipated, given that the prediction model is the first of its kind, and in the study conclusion we encourage other researchers to explore the model's performance across diverse populations.

I found this work to be well written and well explained, with a great deal of thought given to both the value of this work and its current and future limitations.

I would like to thank the authors for their work.

Thank you for recognising the effort and academic considerations invested in this work.

NB : Flow chart is not readable. 

Thank you for drawing our attention to this. We apologise for the poor resolution in the flowchart and have corrected this.

 Reviewer #2: I thank the authors for their work, which I found interesting. Indeed, it would be interesting to have a model that predicts the risk of dystocia and more importantly to predict whether therapeutic measures are efficient (ie. worth the wait and the risk of oxytocin augmentation)

We are grateful to Reviewer 2 for their careful consideration of our manuscript and your positive feedback on the manuscript language and presentation (Q4). 

It was not within the scope of this study to predict the efficiency of treatment by oxytocin augmentation, but it is indeed of interest. As stated in the introduction paragraph of the manuscript, two recent studies have sought to predict the risk of emergency cesarean due to labor dystocia – or in other words the result of inefficient therapeutic measures for treating labor dystocia. We aim for our study to add to this current body of evidence and have therefore targeted the diagnosis of labor dystocia to facilitate a discussion of prevention: “... developing a model that can support risk stratification of women at the onset of labor to target resources and facilitate discussion of prevention.” (L269-270). 

Nevertheless, I feel that several modifications need to be made in the design of the study to assess this important problematic.

First and mostly: the outcome to be predicted.

1. The authors took labour dystocia, without differentiating dystocia during the active phase / descent phase or expulsion phase, which would potentially have different impact, namely a caesarean section is more likely in case of dystocia during the active phase than during the expulsive phase, where an instrumental delivery would be possible. The consequences for maternal morbidity are different.

We fully concur with Reviewer 2 that differentiating dystocia based on the stage of labor could provide a more informative prediction model. Recent research, and our clinical experience, aligns with Reviewer 2’s comment indicating that the mode of birth is influenced by the stage at which labor dystocia is diagnosed . In the original manuscript we point to this the limitation in our discussion: “Neither were we able to subclassify labor dystocia according to the phase of labor.” (L311-312). To clarify that distinguishing between stages of labor would have been relevant, but was a condition of the data source, we have now added the following text: “This distinction could have a clinically relevant implication, as research suggests that the mode of birth is influenced by the timing of labor dystocia diagnosis. Specifically, women diagnosed at ≤5 cm of cervical dilation exhibit higher cesarean section rates [70]. However, obtaining information on the phase of labor from MBR is not possible. While a robust prediction model requires a large population; making the optimal data source a national registry, this approach also poses a limitation due to national registries lacking the additional clinical details sometimes desired.” (L312-318) 

2. they choose to predict all dystocia, without taking into account the measures taken ie. amniotomy or oxytocin augmentation, at least after 2019 there is no information, and therefore I find it difficult to draw conclusions on whether the dystocia was resolved or not, and more importantly whether it was associated with an “adverse” outcome such as caesarean section, instrumental delivery.

Thank you for this comment that seems related to Reviewer 2’s first point. Unfortunately, it was not within the scope of this study to examine the effect of treatment, intrapartum events or take into account whether labor dystocia was resolved. Instead, it attempts model development for the prediction of dystocia to facilitate prevention. 

Second :I find it difficult to understand who was the training dataset and the validating dataset chosen. How many cases of dystocia were in each data set. Figure 1 is unreadable.

We apologise for the poor resolution in the flowchart and have corrected this. With Figure 1 now readable, it should be clear that the training and validation datasets were generated through a random split of the study population and not manually selected. Some readers may not be acquainted with the statistical considerations underlying prediction modelling, and we have therefore added a reference to L147 in the statistical methods section to establish that the approach is substantiated. 

Regarding the distribution of the predicted outcome, ‘labor dystocia’ has been added to Table 2, depicting the study population’s characteristics. This illustrate an almost equal distribution of dystocia cases (44.1% dystocia in the training dataset vs. 44.9% in the validation dataset). 

Third : I feel that the variables testing should be presented as main results. It is indeed the whole point of the study and should not be presented as supplementary data.

We sincerely thank Reviewer 2 for this point, which has been highly helpful in improving the coherence of the manuscript. Our study was two-part. Firstly, we aimed to develop the prediction model and then present the final best-performing model. While model development has received attention in text, as Reviewer 2 states, the related tables should not be supplementary. We now more clearly communicate model development in the manuscript by moving ‘S2 Table’ presenting the eight candidate prediction models to the main manuscript, and it is now Table 1. 

We have considered moving the supplementary table of model input with associated coefficients but are uncertain whether it would contribute significantly to the manuscript. For readers unfamiliar with the statistical differences between association or regression studies and predictive modelling, there might be confusion about the significance of coefficients in selecting candidate predictors. In predictive modelling, the primary concern lies in whether these predictors enhance predictive accuracy rather than demonstrating association. Additionally, the use of restricted cubic splines for the two continuous candidate predictors age and BMI means they are presented twice, which might further complicate interpretation of the table. Due to these considerations, we have decided to retain this table as supplementary for now (S2 Table Selected model input with associated coefficients and standard error). However, if the Editors find it recommendable to include it in the main manuscript, we are willing to do so.

Finally In the end their model seems to have a poor discrimination value…

This point raised by Reviewer 2 is accurate. The final model performed poorly assessed by evaluation measures such as AUC. Because of this, we conclude that the model should be further developed before integration in clinical practice. 

Still, the authors find it important to publish the results of imperfect prediction models as this aligns with transparency and scientific integrity principles. By openly sharing our findings, we not only acknowledge the challenges encountered but also contribute to the collective understanding of predictive modelling in our field. Negative results are valuable contributions, as they inform researchers about what approaches may not be effective and prevent the duplication of efforts. Furthermore, while the described prediction model may not be perfect, it opens opportunities for improvement. Encouraged by Reviewer 2’s comment, we now have unfolded this point in the conclusion: 

The last part of our conclusion previously read: “The prediction ability of the model was satisfactory, and developing a user-friendly online calculator tool, primarily intended for clinicians, is a logical next step. Nevertheless, as with any new model, there is a need for further development, extensive testing, and external validation in other settings to ensure that the model performs well in different populations or clinical settings.”

This has been changed to: “The model's prediction ability was reasonable, but not satisfactory for clinical application. As with any new model, there is a need for further development, extensive testing, and external validation in other settings to ensure that the model performs well in different populations or clinical settings. We hope other researchers will build upon our work - ultimately advancing the field towards more accurate predictive models. Over time, as the development of robust models for predicting labor dystocia progresses, creating a user-friendly online calculator tool, primarily aimed at clinicians, emerges as a logical next step.” (L335-342)

Reviewer #3: To sum up, the study is interesting. A bibliography review shows a few articles on the subject, confirming the review you have already carried out on the subject. The idea of including only predictors present at the onset of labor is very interesting, so we can also screen for them upstream and use them clinically.

Thank you to reviewer 3 for highlighting two central positives of our work. We sincerely believe the novelty of developing a prediction model for the commonly occurring yet enduring complication of labor dystocia holds the interest for many clinicians and researchers alike. Our emphasis on clinical applicability guided our decision to include predictors known at onset of labor. 

The statistical method used in the study seems rigorous and appropriate for developing a predictive model. It's worth noting that the model's predictive performance is currently modest, with a Brier score of 24% and an AUC of 63%. However, this also presents an opportunity for future research to strengthen its predictive capabilities. 

We appreciate Reviewer 3’s positive feedback regarding the manuscript’s provision of data to substantiate the conclusions (Q1) and the application of appropriate and rigorous statistical methods (Q2). Like Reviewer 3 writes, we view our model's modest performance as an opportunity for further refinement and development. While we understand that the initial development of a prediction model rarely provides an immediate 'ready-to-use' tool, we firmly believe in the importance of not only publishing positive outcomes but also detailing initial advances, enabling other researchers to benefit from our progress. Please see the changes to the manuscript's conclusion related to this in extension of your and Reviewer 2’s comments. 

The summary is well written.

It respects the form and content of the article.

Regarding the introduction, it leads to an apparent problem by showing the benefits of prediction 

Thank you. We appreciate the acknowledgement of the clarity and structure of the summary and introduction.

The results of fertility treatments are interesting. Whether they are confirmed or just the result of chance (or a confounding factor) remains to be seen. 

The authors very much agree, and ‘fertility treatment’ as a predictor will be interesting to follow when the model is applied to other populations. 

Q6. PLOS authors have the option to publish the peer review history of their article (what does this mean?). If published, this will include your full peer review and any attached files.

Do you want your identity to be public for this peer review? For information about this choice, including consent withdrawal, please see our Privacy Policy.

Reviewer #1 No

Reviewer #2 No

Reviewer #3 Yes: Buisson Alexandre

---

## [Decision Letter · Decision Letter 1]

16 Jul 2024

Maternal age and body mass index and risk of labor dystocia after spontaneous labor onset among nulliparous women: A clinical prediction model

PONE-D-24-07911R1

Dear Dr. Nathan,

We’re pleased to inform you that your manuscript has been judged scientifically suitable for publication and will be formally accepted for publication once it meets all outstanding technical requirements.

Kind regards,

David Desseauve, MD, MPH, PhD

Academic Editor

PLOS ONE

Additional Editor Comments (optional):

Reviewers' comments:

Reviewer's Responses to Questions

**Comments to the Author**

1. If the authors have adequately addressed your comments raised in a previous round of review and you feel that this manuscript is now acceptable for publication, you may indicate that here to bypass the “Comments to the Author” section, enter your conflict of interest statement in the “Confidential to Editor” section, and submit your "Accept" recommendation.

Reviewer #1: All comments have been addressed

2. Is the manuscript technically sound, and do the data support the conclusions?

Reviewer #1: Yes

3. Has the statistical analysis been performed appropriately and rigorously? 

Reviewer #1: Yes

4. Have the authors made all data underlying the findings in their manuscript fully available?

Reviewer #1: Yes

5. Is the manuscript presented in an intelligible fashion and written in standard English?

Reviewer #1: Yes

6. Review Comments to the Author

Reviewer #1: Thank you for giving me the opportunity to reread this article. I consider that the clarifications I requested have been provided. This work opens up the possibility for other teams to test the model in order to judge its effectiveness.

7. PLOS authors have the option to publish the peer review history of their article (what does this mean?). If published, this will include your full peer review and any attached files.

Reviewer #1: No

---

## [Editor Report · Acceptance letter]

26 Jul 2024

PONE-D-24-07911R1 

PLOS ONE

Dear Dr. Nathan, 

I'm pleased to inform you that your manuscript has been deemed suitable for publication in PLOS ONE. Congratulations! Your manuscript is now being handed over to our production team.

Kind regards, 

on behalf of

Dr. David Desseauve 

Academic Editor

PLOS ONE